# Folding Features and Dynamics of 3D Genome Architecture in Plant Fungal Pathogens

Chongjing Xia,[a,b,f] Liang Huang,[a,c,d] Jie Huang,[b] Hao Zhang,[a] Ying Huang,[e] Moussa Benhamed,[e] Meinan Wang,[f] Xianming Chen,[f,g] Min Zhang,[c] Taiguo Liu,[a,d] Wanquan Chen[a,d]

aState Key Laboratory for Biology of Plant Diseases and Insect Pests, Institute of Plant Protection, Chinese Academy of Agricultural Sciences, Beijing, China
bWheat Research Institute, School of Life Sciences and Engineering, Southwest University of Science and Technology, Mianyang, Sichuan, China
cCollege of Agronomy, Sichuan Agricultural University, Chengdu, Sichuan, China
dNational Agricultural Experimental Station for Plant Protection, Gangu, Ministry of Agriculture and Rural Affairs, Gansu, China
eUniversité de Paris, Institute of Plant Sciences Paris-Saclay (IPS2), Paris, France
fDepartment of Plant Pathology, Washington State University, Pullman, Washington, USA
gU.S. Department of Agriculture, Agricultural Research Service, Wheat Health, Genetics, and Quality Research Unit, Pullman, Washington, USA

Chongjing Xia and Liang Huang contributed equally to this work. Author order was determined by the corresponding author after negotiation.

**ABSTRACT** The folding and dynamics of three-dimensional (3D) genome organization are fundamental for eukaryotes executing genome functions but have been largely unexplored in nonmodel fungi. Using high-throughput sequencing coupled with chromosome conformation capture (Hi-C) data, we generated two chromosome-level assemblies for *Puccinia striiformis* f. sp. *tritici*, a fungus causing stripe rust disease on wheat, for studying 3D genome architectures of plant pathogenic fungi. The chromatin organization of the fungus followed a combination of the fractal globule model and the equilibrium globule model. Surprisingly, chromosome compartmentalization was not detected. Dynamics of 3D genome organization during two developmental stages of *P. striiformis* f. sp. *tritici* indicated that regulation of gene activities might be independent of the changes of genome organization. In addition, chromatin conformation conservation was found to be independent of genome sequence synteny conservation among different fungi. These results highlighted the distinct folding principles of fungal 3D genomes. Our findings should be an important step toward a holistic understanding of the principles and functions of genome architecture across different eukaryotic kingdoms.

**IMPORTANCE** Previously, our understanding of 3D genome architecture has mainly come from model mammals, insects, and plants. However, the organization and regulatory functions of 3D genomes in fungi are largely unknown. In this study, we comprehensively investigated *P. striiformis* f. sp. *tritici*, a plant fungal pathogen, and revealed distinct features of the 3D genome, comparing it with the universal folding feature of 3D genomes in higher eukaryotic organisms. We further suggested that there might be distinct regulatory mechanisms of gene expression that are independent of chromatin organization changes during the developmental stages of this rust fungus. Moreover, we showed that the evolutionary pattern of 3D genomes in this fungus is also different from the cases in mammalian genomes. In addition, the genome assembly pipeline and the generated two chromosome-level genomes will be valuable resources. These results highlighted the unexplored distinct features of 3D genome organization in fungi. Therefore, our study provided complementary knowledge to holistically understand the organization and functions of 3D genomes across different eukaryotes.

**KEYWORDS** 3D genome, Hi-C, plant pathogen, fungi, wheat stripe rust

Address correspondence to Taiguo Liu, liutaiguo@caas.cn, or Wanquan Chen, wqchen@ippcaas.cn.

The authors declare no conflict of interest.

Chromosomes of higher eukaryotes are hierarchically packaged in the nucleus and form three-dimensional (3D) genome organization (1, 2). In mammalian cells, chromosomes are segregated into defined territories at the chromosome scale and are rarely entangled with each other (3, 4). At the megabase scale, chromosomes are further organized into A and B compartments that represent open and closed chromatins, respectively (5, 6). Within the compartments at the submegabase scale are subtle topological domains and loops that function to regulate the interactions between regulatory elements and their physically distal target genes (7, 8). In plant cells, there are two configurations at the chromosome scale, namely, the Rabl and the Rosette (9). In the Rabl configuration, centromeres are clustered at one pole in the nucleus, while telomeres are also clustered but at the other pole. In the Rosette configuration, (peri)centromeric heterochromatins form highly condensed chromocenters, while the euchromatin emanates outward. Similar to mammals, plant cells also have A/B compartments and topological domain-like structures even though they might be formed by different mechanisms (10).

While a plethora of studies have focused on higher eukaryotes, studies on the 3D genome organization in fungi have lagged until the recently developed high-throughput sequencing coupled with chromosome conformation capture (Hi-C) technology (5), for example, in model fungi like budding and fission yeasts (11, 12), *Neurospora* (13, 14), and *Plasmodium* (15, 16). Overall, the chromosomes of these model fungi adopt the Rabl configuration with chromosome territories diminished. Emerging evidence also reveals that a fractal globule (self-interacting chromatin) might be a common feature at a smaller level (~100s of kilobases) (13). Aside from these model fungi, however, a global picture of the 3D genome architecture in nonmodel fungi is not yet available.

While most studies attempted to unravel functional relationships between the 3D genome architecture and transcriptional regulation, differences of the 3D genome organization among species during evolution remain unclear. Earlier studies suggest that the 3D architectures are evolutionarily conserved at various hierarchies (7, 8). However, recent studies reveal a somewhat controversial picture (17). For example, in primates, the lower-order organizations (e.g., specific contacts) are conserved between humans and chimpanzees, but the higher-order structures (e.g., topological domains) are not conserved genome-wide (18). The interspecies differential contacts are associated with the differences of gene expression and, thus, likely have a regulatory function. In *Drosophila* species, a high level of conservation was observed at both compartment and topological domain levels and was maintained over 40 million years even though extensive genome rearrangement occurred (19, 20). However, the dynamics of 3D genome architecture during developmental stages and speciation in fungi remain to be elucidated.

Rust fungi are a group of plant pathogens, taxonomically belonging to the Basidiomycota division, posing persistent threats to global food security (21). Among them are three wheat rust fungi, namely, *Puccinia striiformis* f. sp. *tritici*, *Puccinia graminis* f. sp. *tritici*, and *Puccinia triticina*, causing stripe rust, stem rust, and leaf rust diseases on wheat, respectively, and resulting in significant economic losses worldwide. These three fungi have relatively larger genomes (>80 Mb) than most other pathogenic fungi (22–24), hindering the genomic and epigenomic studies. While sharing common features, e.g., having five different spore stages during their complete lifecycles, needing taxonomically distant plants to complete the sexual stage, and being dikaryotic (unfused two nuclei in single cells, $n+n$) in the urediniospore (the most common stage in nature), they have high specificities in host range under the species level and virulence. Previously, we estimated that these three fungi diverged ~34 million years ago (25), providing an opportunity for studying their 3D genome organization among species during evolution. The objectives of this study were to (i) generate chromosome-level genomes for *P. striiformis* f. sp. *tritici*, (ii) investigate the folding features of the 3D genome architecture in rust fungi, and (iii) unravel the dynamics of the 3D genome organization during development and speciation. This study will underscore the distinct

characteristics of the 3D genome architecture in the Fungi kingdom and provide a basis for further regulatory function studies.

## RESULTS

In this study, we selected two *P. striiformis* f. sp. *tritici* isolates. The first isolate was from China, representing race CYR34 (Chinese yellow rust, race 34; hereafter, CYR34), which is currently the most prevalent race in China. The other one was isolate 93-210, collected from Montana in the United States in 1993 and used in a previous study (25). In the present study, we first assembled these two isolates to the chromosome level using the long reads from PacBio and the short reads from the Illumina HiSeq 2500 PE150 platform, coupled with the Hi-C deep sequencing of genomic DNA from urediniospores. Then, we reconstructed the 3D genome organization of the two isolates using the Hi-C reads. To validate our findings, we also reanalyzed 17 currently available Hi-C data of different fungi. Next, we generated the Hi-C data of CYR34 at the germ tube stage, the structure produced after germination of urediniospores. Coupled with the transcriptomal data, we determined the relationship between the changes of 3D genome organization and the gene expression patterns between these two developmental stages of *P. striiformis* f. sp. *tritici*. We further investigated the chromatin conformation conservation between different fungal species, e.g., *P. striiformis* f. sp. *tritici* versus *P. graminis* f. sp. *tritici* and *Verticillium alfalfae* versus *Verticillium tricorpus*. Finally, we provided an example on how the chromosome-level genome and its 3D organization could lead our understanding of the genomic environment of virulence genes in the plant pathogenic fungi.

**Chromosome-level genome assembly.** The PacBio and Illumina reads were initially assembled into contigs (see Table S1 in the supplemental material), followed by polishing using the Illumina reads (see Table S2 in the supplemental material). Next, we generated the *in situ* Hi-C data for CYR34 (328 million pairs, ~94 Gb, 1,150× sequencing depth) and 93-210 (412 million pairs, ~123 Gb, 1,602× sequencing depth) (see Fig. S1 and Table S1 in the supplemental material). This extremely high-density contact information and extensive manual refinement enabled us to present the final assembly with 18 chromosome-length scaffolds ranging from 2.6 Mb to 5.9 Mb in size (Fig. 1A; see also Table S2).

The CYR34 and 93-210 genomes were predicted to harbor 17,095 and 17,946 protein-coding genes, respectively, of which 9.1 to 9.3% are secreted protein-coding genes (Fig. 1B). A total of 28.77 Mb (36.38%) and 27.89 Mb (35.67%) repetitive sequences were identified in the CYR34 and 93-210 genomes, respectively, of which the long terminal repeat (LTR) retrotransposons (11.05 Mb or 13.97% in the CYR34 genome; 10.27 Mb or 13.14% in the 93-210 genome) and the DNA transposons (9.61 Mb or 12.15% in the CYR34 genome; 10.05 Mb or 12.86% in the 93-210 genome) were the most abundant for both genomes (Table S2). The BUSCO completeness of annotation was 97.0% for both genomes. The Hi-C linking information also enabled us to identify centromeres, with one for each chromosome-length scaffold, for both genomes (Fig. 2A). This, together with well collinearity between our assembled genomes and the *P. graminis* f. sp. *tritici* 21-0 genome of the stem rust fungus, further supported the chromosome-level of the CYR34 and 93-210 genomes (see Fig. S2 in the supplemental material).

**The combination of the fractal globule model and the equilibrium globule model of genome organization in rust fungi.** We observed a large amount of interchromosomal interactions over all valid Hi-C contacts, ranging from 9.70% to 62.85% with an average of 33.05% (see Table S3 in the supplemental material). The prominent "X-shape" structures (Fig. 2A), formed by the interactions among centromeres (26), indicated the colocalization of centromeres in the nucleus (see Fig. S3 in the supplemental material).

In the reconstructed 3D genome structures, each chromosome is tightly interwoven with the remaining chromosomes throughout the nuclear space, instead of spatially restricted to the discrete chromosome territories as in mammalian genomes (Fig. 2B; see also Fig. S3 and Movie S1 in the supplemental material) (2). However, individual

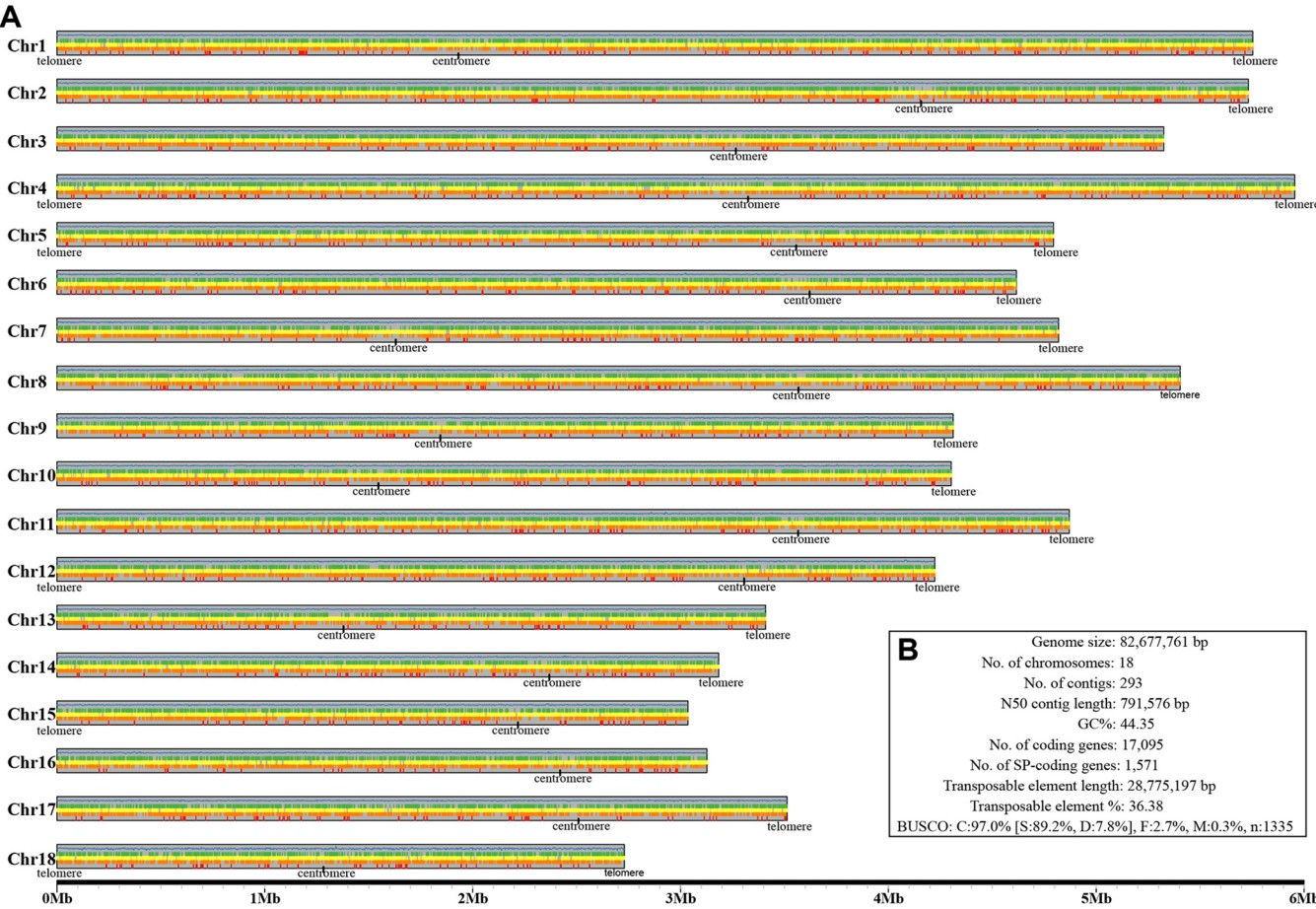

**FIG 1** Chromosome-level genome assembly of CYR34. (A) Plot of 18 chromosomes in CYR34 genome. Five tracks for each chromosome. From top to bottom, distribution of GC%, CpG islands, transposable elements, predicted protein-coding genes, and secreted protein-coding genes. (B) Summary of CYR34 genome assembly.

chromosomes are not randomly dispersed through the nuclear space but loosely entangled themselves to occupy a relatively compact space (see Fig. S4 in the supplemental material). We further observed the clustering of centromeres and telomeres on the opposite sides of the nucleus (Fig. 2C), resembling the canonical Rabl conformation (9, 11). In the *P. striiformis* f. sp. *tritici* 93-210 genome, even though the telomeres are loosely clustered in visualization, the apparent "X-shape" structures in the contact heatmap indicated the strong interactions among centromeres, which is in line with the Rabl conformation. We speculated that the extremely high interchromosomal interactions in the *P. striiformis* f. sp. *tritici* 93-210 genome (62.85% versus the average of 33.05%) might diminish the centromere-centromere interactions when inferring the 3D genome structure. The Rabl conformation was also observed in all fungal genomes that we examined in this study, even though the telomeres are loosely clustered in some cases (Fig. S3).

We then sought to further examine and plot the intrachromosomal contact probability as a function of genomic distance. We calculated the scaling of contact probability ($s$), which should be $s^{-3/2}$ in the equilibrium globule model (5). As expected, the contact probability decreases monotonically as the genomic distance increases from 300 bp to 1 Mb for all samples that we examined (Fig. 2D). Remarkably, we noticed the following two linear fitting slopes in the log-log plot of contact probability: the first one with $s^{-0.83}$–$s^{-1.02}$ at the genomic distance of several to 10s of kilobases, corresponding to the fractal globule model, and the second one with $s^{-1.55}$–$s^{-2.70}$ at the genomic distance of 100s of kilobases, corresponding to the equilibrium globule model (Fig. 2D; see also Fig. S3). This suggested that the nearby loci on fungal genome tend to form monochromatic

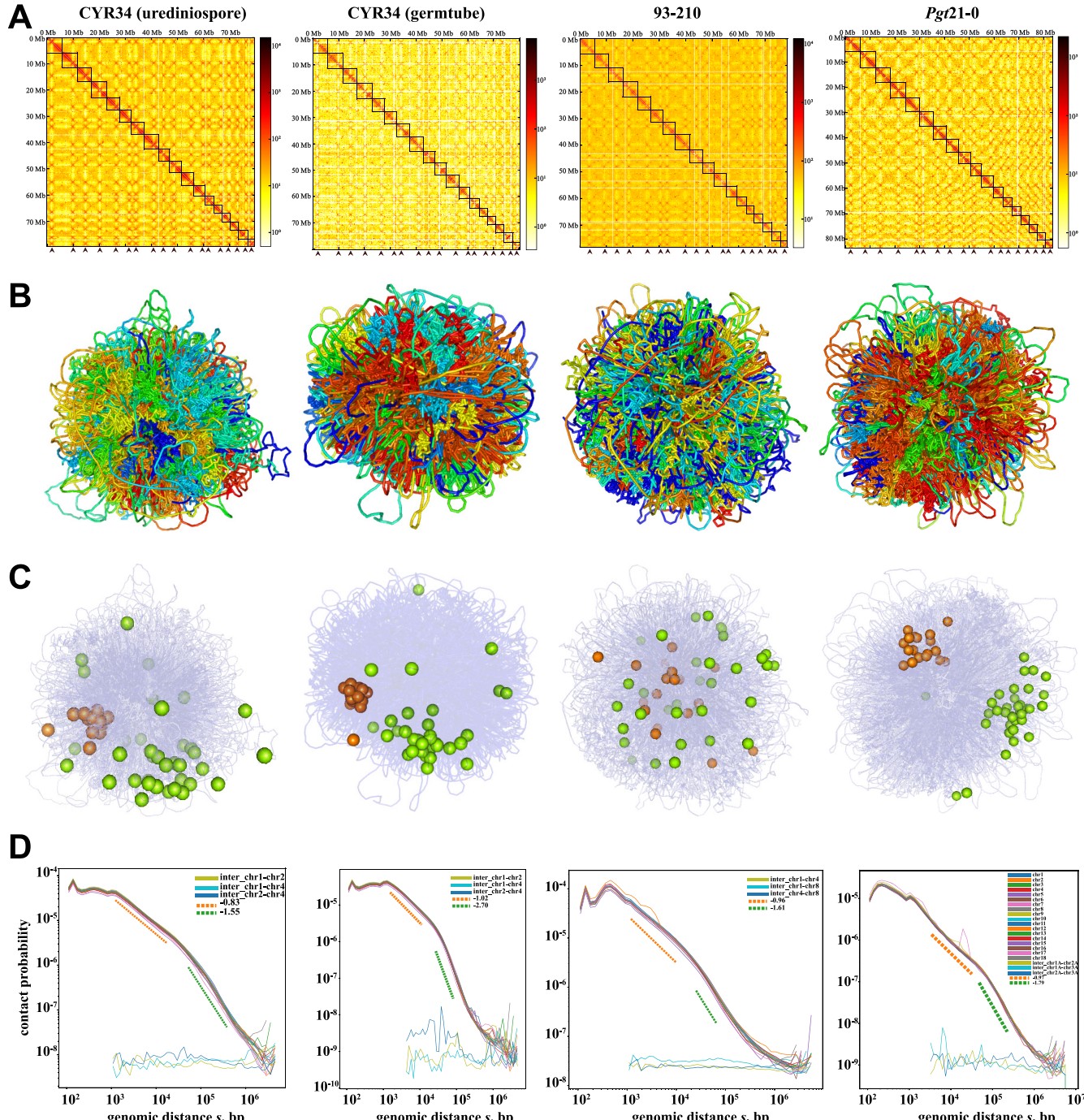

**FIG 2** Combination of the fractal globule model and the equilibrium globule model of genome organization in rust fungi. (A) Contact maps showing frequencies of pairwise 3D genome contacts. Colors represent number of contacts inferred from Hi-C data. Centromeres are highlighted by arrowheads. (B) Reconstructed 3D genomes based on the beads-on-a-string model. Each bead is 500 bp, and each colored string represents one chromosome. (C) Centromere and telomere locations in the reconstructed 3D genomes. Centromeres and telomeres are represented by golden and lemon spheres, respectively. (D) Log-log plot of intra- and interchromosome contact probability $P(s)$ as a function of genomic distance. The dashed lines are linear fitting slopes of the $P(s)$ curves. CYR34 and 93-210 are isolates of *Puccinia striiformis* f. sp. *tritici* (the wheat stripe rust pathogen) and *Pgt*21-0 is an isolate of *P. graminis* f. sp. *tritici* (the wheat stem rust pathogen).

blocks (like fractal globules) in short genomic distances, while chromosomes tend to be highly entangled in large genomic distances (forming equilibrium globules).

We noticed the unusual exponent of $s^{-2.7}$ in the germ tube stage. Given that the germ tube cells are highly active mitotic cells, we speculate what was captured by the Hi-C data was the transient feature of chromosome state from germ-tube cells instead

of chromatin contacts as from the dormant urediniospores. This could be supported by the Hi-C heatmap of germ tubes (Fig. 2A) in which the interchromosome contacts were significantly diminished (9.7% in germ tubes versus 29.93% in urediniospores) (Table S3). We reason that the exponent of $s^{-2.7}$ indicates that the contact probability of two loci decreases sharply as genomic distance increases. In other words, there are less contacts between loci with large distances (>20 kb), which suggests a condenser organization and, therefore, is consistent with the chromosome state.

In summary, we conclude that the 3D genomes of wheat rust fungi have the classical Rabl conformation, and the chromatin organization follows a combination of the fractal globule model in short genomic distances and the equilibrium globule model in large distances.

**The regulation of gene activities independent of the changes of genome organization between the urediniospore and germ tube stages in *P. striiformis* f. sp. *tritici*.** A urediniospore is a relatively dormant stage for the wheat rust fungi to survive the harsh environment, while a germ tube produced from a urediniospore represents an initial stage for growth and interaction with the host. We evaluated dynamics of the chromatin conformational changes between these two stages in CYR34. Across the 18 chromosomes, we identified 396 regions with prominent structural variations between the two stages (Fig. 3A and B), among which 341 regions (totally 20.44 Mb) exhibited clear changes with high confidence (z normalized score of ≤−1.2 and signal-to-noise ratio of ≥0.6). We termed these regions as "dissimilar" regions. We found a linear trend of accumulated length of dissimilar regions over the length of chromosomes (*F* test, *P* = 0.0032) (Fig. 3C), indicating the random distribution of dissimilar regions over chromosomes.

We then attempted to determine whether the dynamics of chromatin conformation plays a regulatory role during stage transition. If so, the change of gene expression in the dissimilar regions might be more dramatic than that in the conservation regions (defined by z normalized score of ≥1.2 and signal-to-noise ratio of ≥0.6). In other words, the expression of upregulated genes should be increased more [higher value of $\log_2$(fold change)] in the dissimilar regions than in the conservation regions. Similarly, the expression of downregulated genes should be decreased more [lower value of $\log_2$(fold change)] in the dissimilar regions than in the conservation regions. Our comparison showed that the fold changes of gene expression in these two regions had no significant difference for overall genes and both upregulated and downregulated genes (Fig. 3D). We further tested the correlation between the gene expression patterns and their 3D genomic locations. Visualization of our reconstructed model of the CYR34 genome suggested that both active and inactive genes are randomly scattered for both the urediniospore and germ tube stages (see Fig. S5a in the supplemental material). Moreover, both up- and downregulated genes showed no propensity for clustering (Fig. S5). These results suggested a limited correlation between gene expression patterns and their 3D genomic locations.

**The chromatin conformation conservation is independent of genome sequence synteny conservation in the examined fungi.** Next, we attempted to investigate the 3D genome architecture changes between different fungal species, namely, *P. striiformis* f. sp. *tritici* (CYR34) and *P. graminis* f. sp. *tritici* (21-0). We first detected 86 large syntenic regions, covering 52.10 Mb and 53.89 Mb in CYR34 and *P. graminis* f. sp. *tritici* 21-0, respectively (see Table S4 in the supplemental material). The 3D genome structure changes were analyzed for these syntenic regions. The CHESS z-scores (measure to quantify similarity) for the syntenic region pairs were not significantly different from those of random pairs (Fig. 4A; *P* = 0.32, *t* test), suggesting that the syntenic regions between CYR34 and *P. graminis* f. sp. *tritici* 21-0 may not share similar 3D chromatin organization (as exemplified in Fig. 4B and C). To further test this, we plotted the z-scores over the syntenic scores. As illustrated in Fig. 4D, the correlation between the 3D chromatin changes and the synteny relationship was low. The comparison of these two fungal pathogens with *Verticillium* species (see Materials and Methods; see also Table S4 in the supplemental material) revealed a similar pattern (see Fig. S6 in the supplemental

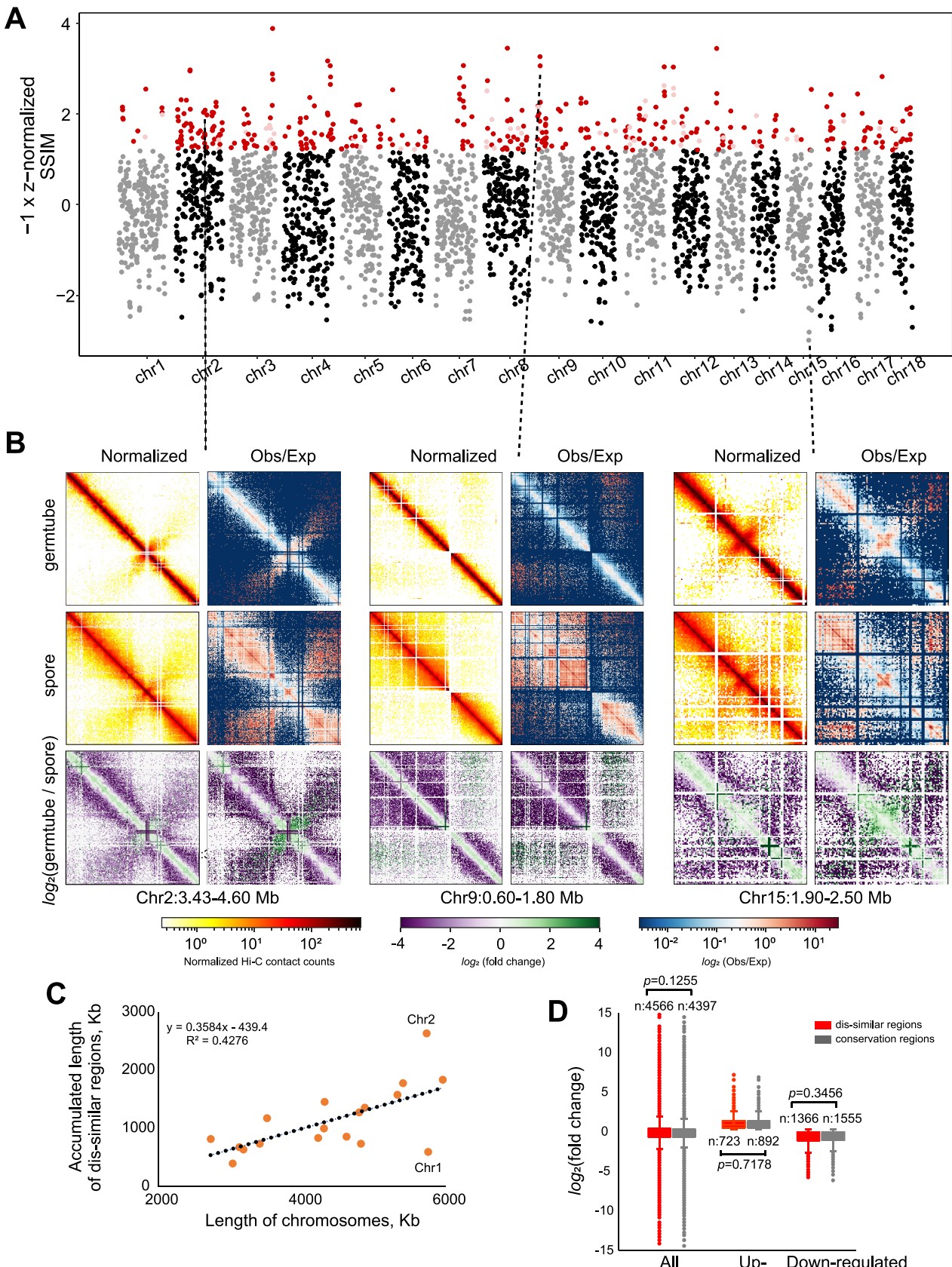

**FIG 3** Regulation of gene activities independent of the changes of genome organization between the urediniospore and germ tube stages. (A) The plot of z normalized similarity score of Hi-C data generated from urediniospores and germ tubes. Each dot represents a 100-kb window. Red dots

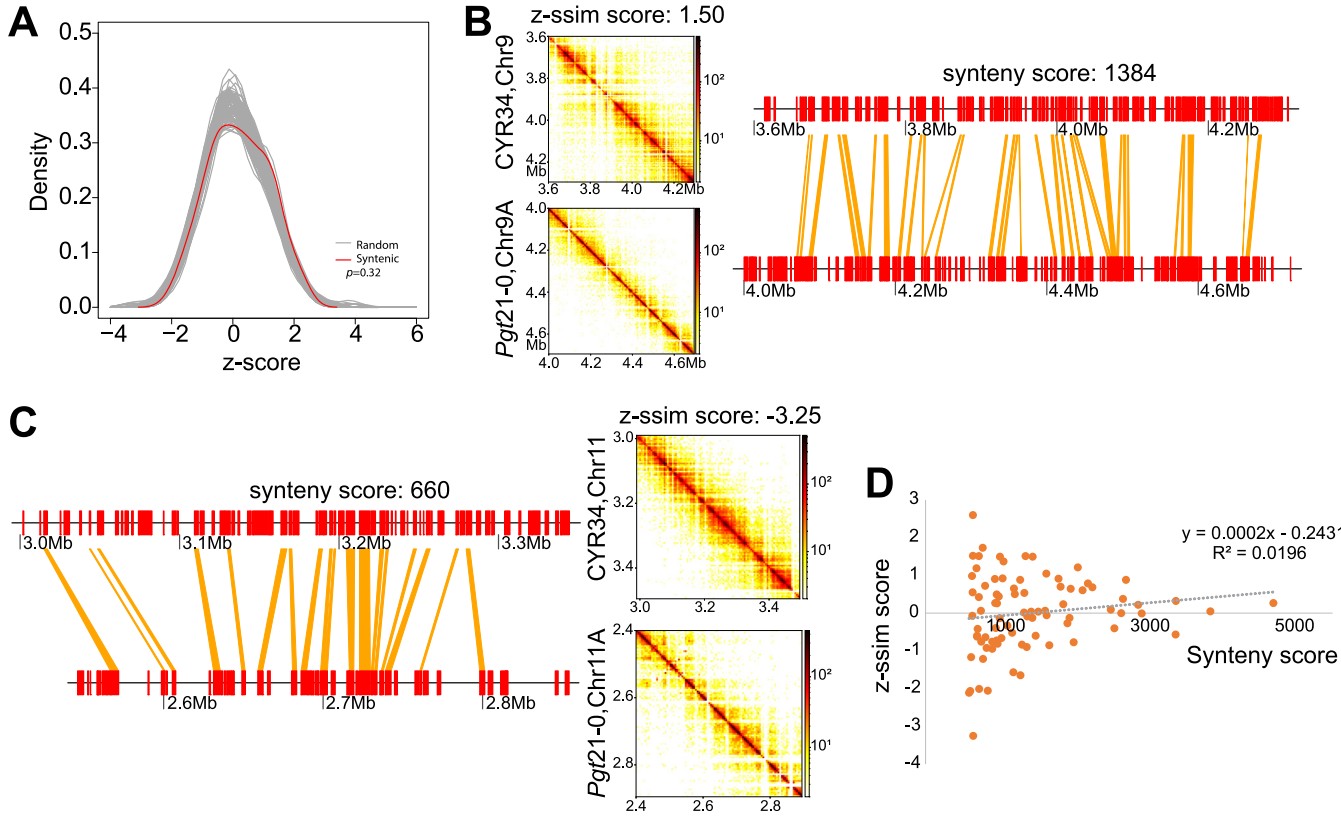

**FIG 4** Chromatin conformation conservation independent of genome sequence synteny conservation in fungi. (A) Distributions of empirically determined CHESS z-scores for syntenic region pairs in CYR34 and *P. graminis* f. sp. *tritici* 21-0 (red line) and 100 random permutations of region pairs (gray lines). (B and C) Two examples of syntenic regions with different z-scores. (D) Distribution of z-scores over syntenic scores. Note the low Pearson correlation coefficient.

material). This suggested that the 3D chromatin conformation conservation is independent of the genome sequence synteny conservation in the filamentous fungi, which is distinct from the cases in mammalian genomes in which the syntenic genome regions share a highly similar 3D chromatin organization (27). Further studies will be needed to examine this independence among diverse fungi when their high-quality Hi-C data are available.

**Compartments were not found in *P. striiformis* f. sp. *tritici* and other fungi.** The contact heatmaps of the syntenic regions between species did not show apparent chromatin organizational patterns (e.g., A/B compartments), which may contribute to the low conservation of chromatin conformation in the syntenic regions. To test this hypothesis, we examined whether A/B compartments exist in the wheat rust fungi. We particularly examined the following: (i) whether the PC+ regions (regions with positive principal component eigenvector value) have significantly higher gene densities and expression levels than the PC− regions (negative eigenvector value); and (ii) the Hi-C contact heatmaps have plaid-like patterns, and whether PC+ and PC− regions are corresponding to these patterns.

Even though the CYR34 chromosomes could be separated into PC+ and PC− regions, the differences (in terms of gene density and expression level) were not statistically significant for almost all chromosomes and all of the first three PCs (Fig. 5A to C). This result was in contrast with the case in the human genome in which both gene

**FIG 3** Legend (Continued)

are highly dissimilar regions (z normalized similarity score of ≤−1.2), while light red dots are noisy regions (signal-to-noise ratio of <0.6). (B) Examples of two chromosomal regions with conformational changes (left and middle) and one region with conformational conservation (right). (C) Distribution of accumulated length of dissimilar regions over chromosomes. Dashed line represents linear modeled regression. (D) Comparison of expression fold changes of genes between highly dissimilar regions and conservation regions.

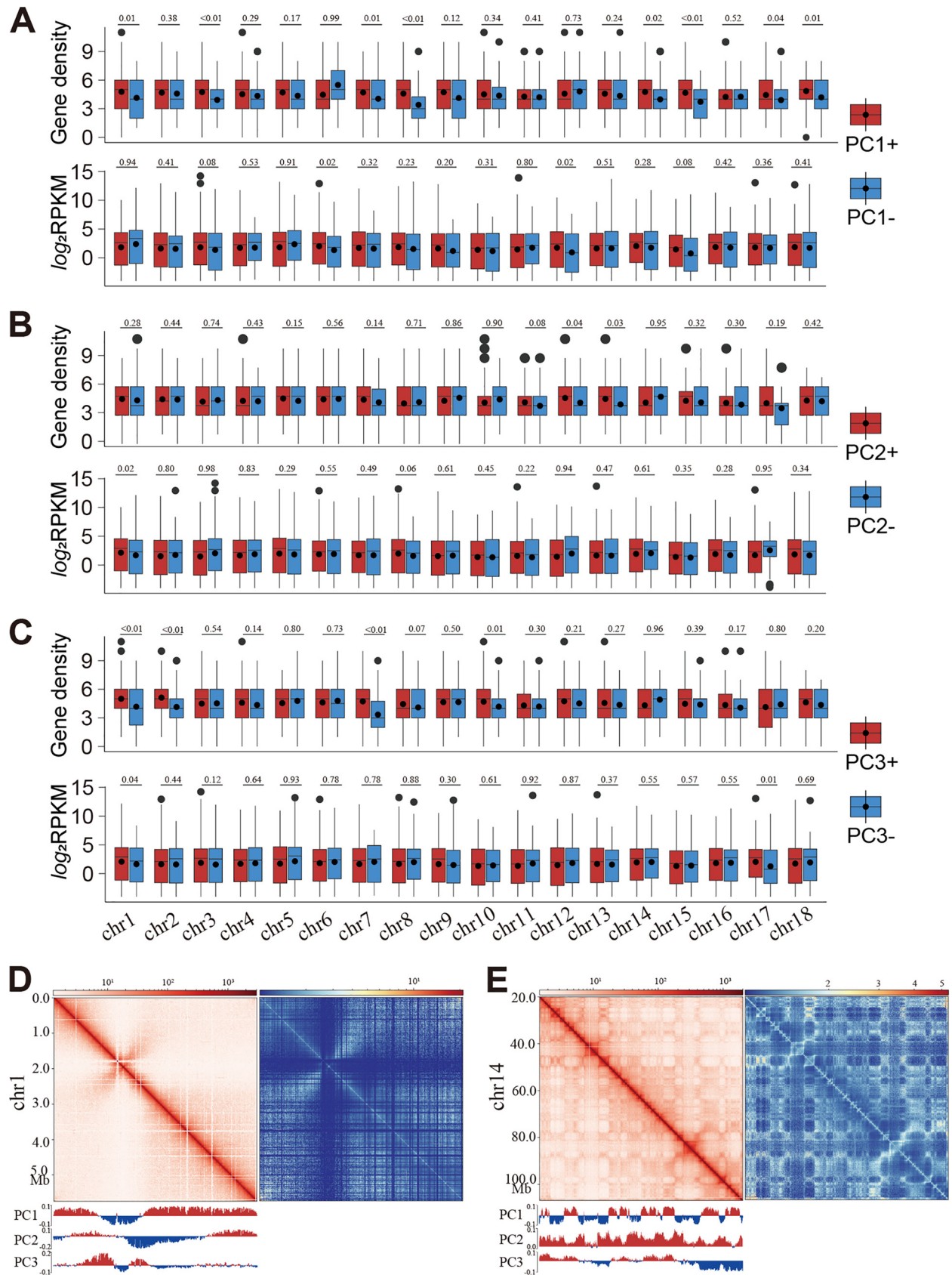

**FIG 5** Compartments not existing in wheat rust fungi. (A to C) Boxplots showing the gene density and gene expression in the PC+ and PC− regions in each chromosome. The PC "+" and "−" regions were designated according to the gene density in each region. The black lines and

density and expression level were significantly higher in the PC+ regions than those in the PC− regions (see Fig. S7 in the supplemental material).

The Hi-C contact heatmaps did not show the plaid-like patterns in the CYR34 genome (Fig. 5D). However, the plaid-like patterns were apparent in the human genomes (Fig. 5E), and the sharp transitions from compartment to compartment exactly corresponded to the values of PC eigenvectors (PC1 for chr14 in Fig. 5E). To further validate our observation in *P. striiformis* f. sp. *tritici* CYR34, we plotted and visually examined the contact heatmaps and the first three PC eigenvectors of 14 fungi (see Fig. S8 in the supplemental material). Most of the examined fungi (13 out of 14) displayed the same pattern as observed in CYR34. *Epichloe festucae* was the only exception, as it showed similar transition patterns of compartment and PC1 values with the human genome (Fig. S8h). In fact, it was previously reported that the compartments divided by PC1 corresponded to high AT (PC1 "+") and GC (PC1 "−") content regions (28). However, such a bipartition of chromosomes by AT content was not observed in *P. striiformis* f. sp. *tritici* (Fig. 1A).

In most cases, PC1 (or PC2 in other cases) separated a whole chromosome into three regions as follows: a first PC "+" region in one terminus of the chromosome, followed by one PC "−" region in the middle and the second PC "+" region in the other terminus (see Fig. S9 in the supplemental material). Remarkably, the centromeres (X-shape in the contact heatmaps) all resided in the PC "−" region. Therefore, we speculated that the PC1 "+" and "−" regions might correspond to the euchromatin and the heterochromatin, respectively. To test this hypothesis, we examined the expression pattern in the PC "−" regions. As expected, the centromere regions within the PC "−" regions were perfectly associated with low-gene-expression levels, indicating that these regions are inactive heterochromatin. The colocalization of H3K9me3 (a heterochromatin-specific histone modification) with the centromeres within the PC "−" regions in *Neurospora crassa* (see Fig. S10 in the supplemental material) further confirmed the heterochromatin nature of the PC "−" regions.

These results suggested that compartmentalization of chromosomes does not exist in the 3D genome folding of the wheat rust fungi or in most other filamentous fungi, but the chromosomes are separated into the euchromatin and the pericentromeric heterochromatin at the megabase scale, which is a distinct feature in comparison with high eukaryotic mammalian and plant genomes.

**Avirulence gene cluster *AvYr44-AvYr7-AvYr43-AvYrExp2* in the *P. striiformis* f. sp. *tritici* 3D genome.** Avirulence genes in plant pathogens are genes encoding products needed to interact with host resistance gene products, and therefore they determine host resistance specificity. Previously, we identified a genome region harboring four avirulence genes in the *P. striiformis* f. sp. *tritici* genome, named the *AvYr44-AvYr7-AvYr43-AvYrExp2* gene cluster (29). Due to the fragmented assembly, we were unable to decipher the detailed genome environment of the avirulence gene cluster. In the present study, we reanalyzed our previously established selfing population and clearly mapped this gene cluster to a subtelomeric region of the short arm in chromosome 7 (Chr7) (Fig. 6). Subtelomeric regions in plant fungal pathogens are usually unstable and contribute to virulence diversity (30), suggesting that such plastic regions may also promote virulence evolving in *P. striiformis* f. sp. *tritici*.

## DISCUSSION

The two chromosome-level genomes of *P. striiformis* f. sp. *tritici*, together with the data of other wheat rust fungi (22, 24, 31), demonstrated the power of incorporating the Hi-C data into scaffolding of large fungal genomes. Previously, we and others were unable to assemble *P. striiformis* f. sp. *tritici* genomes to the chromosome-level, even

**FIG 5** Legend (Continued)

solid circles within boxplots are median and mean; the solid circles above the boxplots are outliers. The numbers above each pair of boxplots represent the *P* values of significance (one-tail Welch *t* test). (D and E) Original and observed/expected contact map of chr1 in *Puccinia striiformis* f. sp. *tritici* CYR34 (D) and chr14 in human Hapl-A cells (E). Below the original contact maps are the distributions of three eigenvector values.

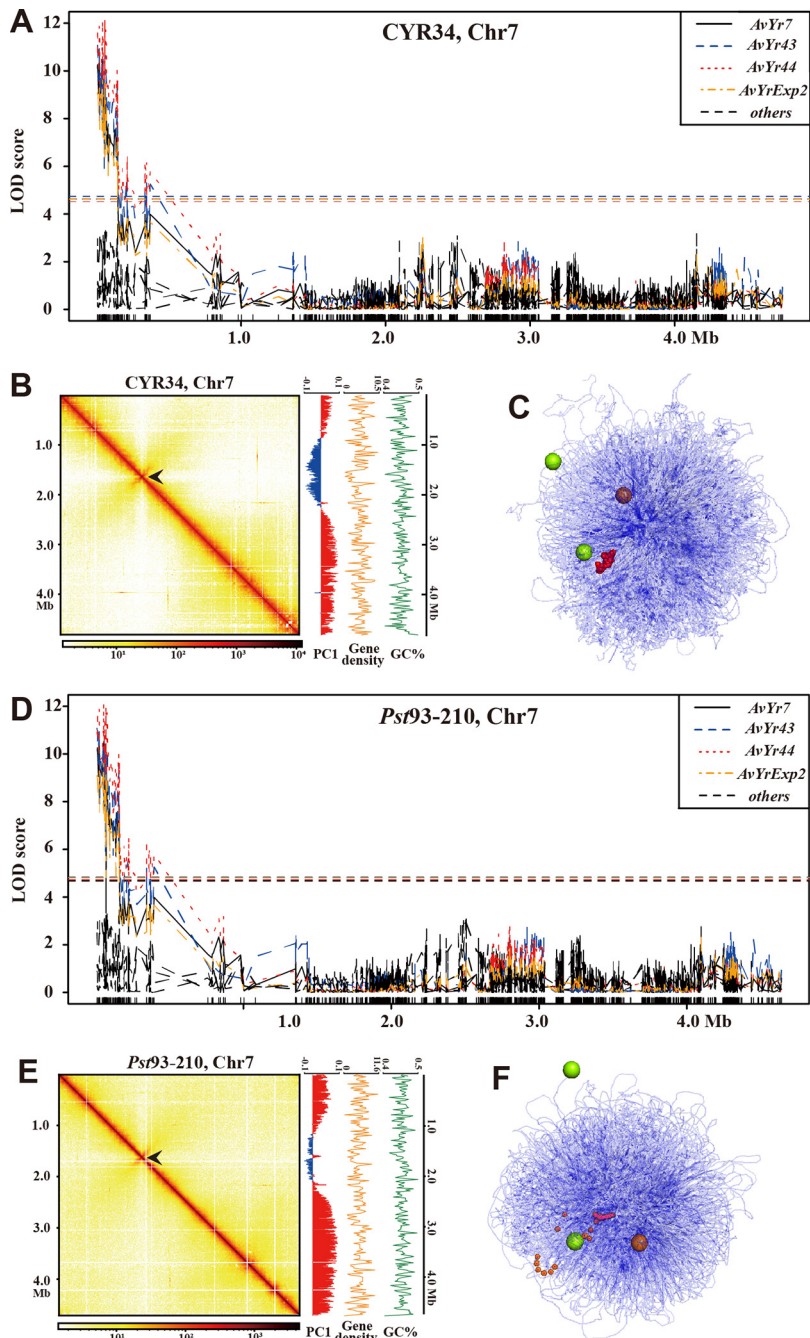

**FIG 6** Avirulence gene cluster in the *P. striiformis* f. sp. *tritici* 3D genome. (A and D) QTL analysis of previously established *Puccinia striiformis* f. sp. *tritici* selfing population mapped avirulence gene cluster *AvYr44-AvYr7-AvYr43-AvYrExp2* to the telomere region of Chr7 in the CYR34 and 93-210 genomes. The color-dashed horizontal lines are the LOD thresholds for different phenotypes. (B and E) The genome architectures of Chr7 in CYR34 and 93-210, respectively. (Left) Heatmap of Hi-C contacts; the color bar is scaled to the contact density. Note that the canonical "X" shape is the centromere region (the arrow), and no topological associated domains are presented. (Right) Plot of PC1 of Hi-C matrix, gene density (number of genes per 20 kb), and GC content of Chr7. (C and F) The *AvYr44-AvYr7-AvYr43-AvYrExp2* cluster in the constructed 3D genome. The red segments are the *AvYr44-AvYr7-AvYr43-AvYrExp2* cluster; and the centromeres and telomeres are represented by golden and lemon spheres, respectively.

using long-read sequencing technologies, mainly due to the relatively large genome of *P. striiformis* f. sp. *tritici* (refer to Table S3 in the supplemental material for a brief comparison) and high percentages of repetitive sequences (23, 32). One advantage of using the Hi-C data is the proximity information between long-range genomic loci

(from 100s of kilobases to several megabases), which could be utilized to anchor, order, and orient initial contigs to chromosomes (33). The Hi-C-assisted scaffolding pipeline used in our study will also be feasible for studying other filamentous fungi with large and complex genomes. The two chromosome-level genome assemblies of *P. striiformis* f. sp. *tritici* generated in this study are valuable resources for further functional and evolutionary genomic studies and, indeed, have provided unprecedent insights into the 3D genome architectures of plant fungal pathogens as discussed below.

Overall, the plant fungal pathogens analyzed in this study have a Rabl chromosome configuration, which was also observed in model fungi (11, 13, 14), suggesting that this configuration might be a conserved feature at the chromosome-scale in the kingdom Fungi. Notably, the Rabl chromosome configuration has also been observed in other organisms with different genome sizes, including insects such as mosquitoes (genome size, 1.2 Gb) (34), maize (2.4 Gb) (10), and plants such as barley (4.79 Gb) (35). This suggested that the formation of the Rabl configuration is irrespective of chromosome length. Recently, Hoencamp et al. (34) demonstrated that the Rabl-like features are determined by the absence of condensin II subunits that promote chromosome compaction during mitosis, and the loss of lengthwise compaction is responsible for clustering of centromeres. However, which and how condensin II subunits (CAP-H2, CAP-G2, and/or CAP-D3) were lost during fungi evolution remain unclear since the Rabl configuration was observed for all fungi examined in our study. Moreover, we noticed that the chromosome conformation of rust fungi also follows a combination of the fractal globule model (in short genomic distances) and the equilibrium globule model (in large distances). This cannot be explained by the loss of condensin II subunits since its deficiency did not affect the finer scale of the 3D genome organization including the intrachromosomal contact probability (34). However, the functional implications of such a distinct genome conformation with combined fractal and equilibrium globule models in fungi remain open questions.

Another distinct feature in the 3D genome organization of plant pathogenic fungi is that the chromosome compartmentalization does not exist. First described in a human lymphoblastoid cell line (5), the compartments correlated well with the transcriptionally active and inactive regions and are therefore called A and B compartments, respectively. The mechanism underlaying the formation of compartments in the genomes of eukaryotes remains unclear. Given that the length of compartments is usually at the several megabase scale and that the length of most fungal chromosomes is <10 Mb, one might speculate that the short length of fungal chromosomes (<10 Mb, compared to >100 Mb for most mammals) could contribute to the absence of compartments. However, this could be easily rebutted by the case in archaeal *Sulfolobus* species that display the presence of two compartments in their single circular chromosome with size ranging from 2.2 to 3 Mb (36). Surprisingly, even in the archaeal kingdom, the compartmentalization was not observed in all species, e.g., *Haloferax volcanii* (37). In our study, we found that the endophytic fungus *Epichloe festucae* was the only exception (out of the 15 tested fungi) (see Fig. S8 in the supplemental material) (38) with the absence of compartments. Our further analysis did not detect significant differences in the number of SMC (structural maintenance of chromosomes) family proteins (see Table S5 in the supplemental material), the proposed key players in the genome organization via "loop-extrusion." Here, we could only provide some speculations on the possible reasons underlying the absence of compartments in most fungi but present in certain fungi. Given that fungi lack cohesin, the observed organization could not be mediated by loop extrusion which needs cohesin. The possible explanation for the observed plaid-pad patterns might be that these patterns are compartment domains formed by interactions of sequences in the same transcriptional state (2). The lack of compartment might be because large segments of their genome are in the same transcriptional state, which is consistent with our observations in *P. striiformis* f. sp. *tritici* (CYR34) and *N. crassa* (see Fig. S9 and S10 in the supplemental material). It would be needed to test this hypothesis by studying the

contribution of different transcriptional states to the compartment domain formation in diverse fungi.

Regardless of its formation mechanism, the presence and absence of the chromosome compartmentalization might exist across organisms in a kingdom. Instead of forming compartments, we speculate that the chromosomes of plant pathogenic fungi might follow a "euchromatin-heterochromatin-euchromatin" (EHE) organization (Fig. S8), given that the centromere regions picked up by eigenvectors are enriched in heterochromatin-associated H3K9me3 modification (Fig. S10). Such an organization has also been observed in fungal pathogens of human and animals (16). We noticed that the heterochromatin regions corresponded well to the AT-rich regions for fungi with a bipartite structure of skewed AT content regions, such as in *Verticillium* and *Neurospora* spp. (Fig. S10). However, other fungi including *Puccinia* spp. displayed a relative equilibrium of the GC verse AT contents along the genome, and therefore, the heterochromatin formation could not be driven only by nucleotide composition. In fact, regardless of the nucleotide composition, we observed that genes in the pericentromeric heterochromatin regions had low expression. This enabled us to speculate that the transcription appears to be associated with the EHE organization. This speculation has also been proposed for chromatin compartments formation, in which the formation of the compartment is mediated by homotypic interactions among genome regions with the same transcriptional state, and then generated phase-separated structures (2). From this perspective, the organization of fungal chromosomes follows the same folding principles as in higher eukaryotes at the megabase scale.

Surprisingly, the changes of 3D genome structure were not associated with the changes of gene expression in the two stages of *P. striiformis* f. sp. *tritici*, which is contrary to the mammalian genomes. While whether this phenomenon was also held in other plant pathogenic fungi remains to be explored, we reasoned that this might be a contributor to the lack of finer genomic organizational structures (e.g., topologically associated domains or loops). In fact, increasing evidence in higher eukaryotes suggests that the chromatin structures act as modulators of transcription activities instead of as deterministic factors (2, 39). Therefore, it is also possible that the changes of chromatin structures only have minor effects on gene transcription activities. Further studies on changes of the 3D genome organization of plant pathogenic fungi on chemical transcription inhibition will be helpful for answering this question.

## MATERIALS AND METHODS

**Urediniospore multiplication.** Two *P. striiformis* f. sp. *tritici* isolates, CYR34 and 93-210, were purified, and urediniospores were multiplicated by inoculating seedlings of wheat variety Chinese 166 for CYR34 and Nugaines for 93-210 using urediniospores from a single uredinium, following the previously described procedure (40). To produce germ tubes, fresh urediniospores were suspended in sterile distilled water and incubated at 10°C for 10 h. Germination rate was estimated by counting germinated spores under a microscope. Germ tubes were harvested for subsequent RNA and Hi-C sequencing when germination rate reached 90%.

**Sequencing.** For Illumina and PacBio sequencing, genomic DNA was extracted from fresh urediniospores following the cetyltrimethylammonium bromide (CTAB) method with modifications as previously described (25). Libraries were prepared using the TruSeq DNA PCR-free library prep kit (Illumina, San Diego, CA, USA) and SMRTbell (Sage Science, MA, USA), following the manufacturer's recommendations, with a 200-bp insertion size for Illumina sequencing and 20-kb fragments selected for PacBio sequencing. For transcriptomic sequencing, total RNA from CYR34 urediniospores and germ tubes were extracted using the TRIzol reagent (Invitrogen, CA, USA). Quality and quantity of extracted RNA were checked using gel electrophoresis and an Agilent 2100 Bioanalyzer (Agilent Technologies, UK). Sequencing libraries were prepared using the Illumina TruSeq Stranded mRNA library prep kit (Illumina, CA, USA). Illumina and transcriptome sequencings were performed using the Illumina HiSeq2500 PE150 platform. Transcriptome sequencing was conducted with cDNA samples from CYR34 urediniospores and germ tubes, and 3 replicate libraries were sequenced for each sample. For PacBio sequencing, 6 libraries and 9 libraries were sequenced using the PacBio Sequel and RS II platforms, respectively. The PacBio reads from these two platforms were merged for subsequent analyses.

*In situ* Hi-C experiments were performed for CYR34 urediniospores and germ tubes and 93-210 urediniospores. For cross-linking of cells, 1g urediniospores or germ tubes was suspended and fixed in 40 mL 3% formaldehyde and incubated for 20 min at 25°C with 200 rpm shaking. Crosslinking was quenched by adding 6.48 mL 2.5 M glycine and incubated for 5 min at 25°C with 200 rpm shaking. For cell lysis, the cross-linked cells were spun down for 5 min at 4°C at 1,800 rpm. The supernatant was

discarded, and 5 mL 1× CutSmart buffer (no. B7204V; NEB, Ipswich, MA, USA) was added to the tube. The resuspended pellet was frozen in liquid nitrogen in a mortar. The frozen cells were lysed by pestle grinding. Five milliliters 1× CutSmart buffer was added to the lysed cells and stored at −80°C for subsequent procedures. Chromatin was digested using restriction enzyme MboI. The Hi-C libraries with two replicates per sample were prepared in Sinobiocore Biological Technology Co. (Beijing, China) and sequenced using an Illumina HiSeq 2500 PE150 platform.

**Genome assembly and annotation.** Before genome assembly, we used Jellyfish (v2.2.10) (41) and GenomeScope (v2.0) (42) to estimate the genome size and heterozygosity. The CYR34 genome had a haploid length of 82,441,790 bp with a heterozygosity of 1.46%, while 93-210 had a haploid length of 77,842,539 bp with a heterozygosity of 1.41%.

For genome assembly, we tested 10 commonly used long-read-based assemblers. The assembly results varied significantly (data not shown). After testing, we found MaSuRCA (43), a hybrid approach using a combination of PacBio and Illumina reads, generated the most contiguous and the highest number of BUSCO hits for the genomes and therefore used for the genome analyses. For both CYR34 and 93-210, MaSuRCA v3.3.5 was used for initial assembly, with default parameters except that the estimated genome size was set accordingly. Then software POLCA within the MaSuRCA package was used for assembly polishing. In this step, 2,353 substitution and 5,936 insertion/deletion errors were corrected, with a final consensus quality of 99.9916. Since the *P. striiformis* f. sp. *tritici* genome is highly heterozygous, we used purge_haplotigs (44) (v1.1.0) to remove putative allelic regions that are collapsed in the MaSuRCA assembly to generate the haploid genome assembly. After allelic regions were purged, an extra step "purge_haplotigs clip" was run, as recommended, to identify and trim the overlapping ends between contigs to generate the final haploid genome. After we generate the haploid genome assembly, we noticed that some of the telomere repeats (TTAGGG and CCCTAAA) were assembled to the middle of the contigs. For this case, we split the contig into two. After this process, the assembly was subject to the following scaffolding steps.

The two Hi-C libraries constructed from urediniospores for each of CYR34 and 93-210 were merged and used for scaffolding the CYR34 and 93-210 genomes, respectively. For scaffolding, we followed the procedure described in https://github.com/theaidenlab/Genome-Assembly-Cookbook. First, Juicer (v1.5.6) (45) was used to process Hi-C reads. In this step, a contact matrix, from which 3D proximity relationships between short contigs could be inferred, was generated and then subject to the analysis using software 3D-DNA (v180922) (33) to correct the misassembles and anchor, order, and orient the contigs into scaffolds. For 3D-DNA, we ran the steps in the order of "run-asm-pipeline.sh –mode diploid –input 10000 –editor-coarse-resolution 2500000 –editor-coarse-region 7500000 –editor-repeat-coverage 4 –polisher-input-size 10000 – polisher-coarse-resolution 100000 haploid.fasta merged_nodups.txt." Minor assembly errors were further detected and corrected according to the contact frequency, and chromosome boundaries were redefined when necessary, using the Juicebox assembly tools (v1.11.08) (45). The assembled genomes of CYR34 and 93-210 were aligned to the *P. graminis* f. sp. *tritici* 21-0 genome using progressiveMauve (v20150226) (46), and the chromosomes of the *P. striiformis* f. sp. *tritici* isolates were named in a way such that chr1 of *P. striiformis* f. sp. *tritici* is homologous to chr1 of *P. graminis* f. sp. *tritici*.

The fungal genome annotation pipeline, FunGAP (v1.1.0) (47), was performed for gene prediction. To assist gene prediction, 11 previously available proteomes (from *Puccinia* spp. and *Uromyces* spp.) were retrieved for the FunGAP analysis. These 11 proteomes are the following: GCA_008522505.1, GCA_008520325.1, GCA_007896445.1, GCA_004348175.1, GCA_004194325.1, GCA_002994595.1, GCA_002994575.1, GCA_002994555.1, GCA_002920065.1, GCA_002920205.1, and GCA_002900275.1 in the NCBI database. The previously trained *P. striiformis* f. sp. *tritici* gene model (32) was selected for the Augustus analysis. Moreover, InterProScan (v5.40-77.0) (48) was used to infer the putative functions of the predicted genes. To predict secreted proteins, the presence of signal peptides was detected using SignalP (v5.0b) (49), and proteins with transmembrane helices were identified and filtered using TMHMM (v2.0c) (https://services.healthtech.dtu.dk/service.php?TMHMM-2.0).

**Hi-C data processing.** We first validated the reproducibility of the Hi-C libraries. For this, the Pearson correlation coefficients were calculated for each pair of two libraries using the hicCorrelate function in HiCExplorer (v3.5.1) (50). For each Hi-C library (see Table S1 in the supplemental material), the trimmed paired-end reads were mapped to our assembled CYR34 genome using bwa-mem2 (51). Then, the HiCExplorer pipeline was followed to process the Hi-C reads. Raw contact matrices were generated initially at the binning resolution of 1 kb and then merged to different bin sizes for subsequent analyses. For Pearson correlation analysis, the raw contact matrices were merged to 50 kb by setting –numBins 50. Function hicCorrelate was used to calculate spearman correlation coefficients, and pairwise scatterplots between libraries for each sample were plotted for visualization.

**Centromere identification.** The Hi-C paired reads were preprocessed and quality control checked using HiCExplorer as described in the Hi-C data processing section. For CYR34 and 93-210, the Hi-C reads were mapped to our assembled genome, the contact interaction was estimated at the resolution of 50 Kb. For *P. graminis* f. sp. *tritici* 21-0, the Hi-C reads were downloaded from NCBI SRA under accession number SRR9024806 (22) and were mapped to the previously published genome, which was retrieved from https://ftp.ncbi.nlm.nih.gov/genomes/all/GCA/008/522/505/GCA_008522505.1_Pgt_210/GCA_008522505.1_Pgt_210_genomic.fna.gz (22 April 2020). Since the validated contacts were fewer than those of CYR34 or 93-210, we used a resolution of 100 Kb for *P. graminis* f. sp. *tritici* 21-0. Software Centurion v0.1.0 (52) was used for centromere identification following the documentation in http://projects.cbio.mines-paristech.fr/centurion/auto_examples/plot_finding_centromeres.html#sphx-glr-auto-examples-plot-finding-centromeres-py. For the *N. crassa* genome (NCBI GCF_000182925.2), the supercont12.6_20 in chromosome VI was inversed, as noted in the original paper (13). So, this sequence was complementarily reversed for all analyses in our study.

**Intrachromosomal and interchromosomal contact probability.** The processed reads in the ".bam" format from the HiCExplorer hicBuildMatrix module were converted to pairs using bedtools, the genome was divided into 1-kb bins, and probability $P(s)$ was calculated using cooltools (v0.3.2) following the documentation in https://cooltools.readthedocs.io/en/latest/notebooks/contacts_vs_distance.html. Briefly, $P(s)$ was calculated through dividing the number of observed interactions in each bin by the total number of possible pairs. To calculate the interchromosomal contact probability, we combined the two chromosomes as a super chromosome. Then the contact probability was calculated between the two loci, one from one chromosome and one from the other. For interchromosomal contacts, $P(s)$ was only calculated among the three longest chromosomes, as shown in Fig. 2D.

**Three-dimensional genome structure reconstruction.** For this analysis, the trimmed and filtered Hi-C paired reads were processed using the nuc_processing (53), with parameters set as follows: -re1 MboI -n 20 -r 3 -o cyr34 -or report -b/path/to/bowtie2 -p -k -sam -v. The 3D genome structures were reconstructed using nuc_dynamics, with parameter set as follows: -s 1 0.5 0.1 0.05 0.01 0.005 0.002 0.001. When constructing the 3D genome, we used the root mean square deviation (RMSD) as an indicator to evaluate the consistency of the structures in the ensemble following the developer's recommendation. For the structures with an RMSD of >2.0, we presented the lower resolution with an RMSD of <2.0. The generated 3D genome structure in the pdb format was visualized in Open-Source PyMOL (Schrödinger LLC., New York, NY) (54).

**Detecting structure and gene expression changes between the CYR34 urediniospore and germ tube stages.** Since the Hi-C data from CYR34 urediniospores had a much larger number of reads and valid contacts, we randomly resampled a fraction of reads (106,358,228 pairs) to make the number of valid contacts (29,738,511) compatible to that of the germ tube library (27,325,134). Then, the contact matrices were binned to 5 kb. The reference genome was segmented into a windows span of 100 kb with a step size of 20 kb. Software CHESS (v0.3.6) (27) was used to compare the two chromatin contact matrices. For differential analyses of gene expression, the transcriptome sequencing data of CYR34 urediniospores and germ tubes were firstly mapped to our assembled genome using HISAT2 (v2.2.0) (55). The R package "zigzag" (v0.1.0) (56) was used to determine whether the genes are actively expressed. The genes with probabilities higher than 0.9 were considered as active or otherwise inactive. Software EmpiReS (v1.0) (57) was used for the differential analysis of gene expression.

**Comparison of syntenic regions between CYR34 and *P. graminis* f. sp. *tritici* 21-0.** The syntenic regions between CYR34 and *P. graminis* f. sp. *tritici* 21-0 (GCA_008522505.1) were identified using MCscan (v0.8) (58). In total, we identified 174 syntenic regions between CYR34 and *P. graminis* f. sp. *tritici* 21-0, from which 86 large regions (harboring >15 genes) were selected for subsequent analysis. To make the Hi-C of CYR34 comparable to that of *P. graminis* f. sp. *tritici* 21-0 (11,550,650 pairs of valid contacts), we resampled 38,198,015 pairs to get 12,126,745 pairs of Hi-C valid contacts. Then, the Hi-C matrices were binned to the 10-kb resolution and were then subject to comparison using CHESS. Following the previous procedure (27), we also performed comparisons between region pairs with shuffled syntenic region IDs as control. This was repeated 100 times. Examples of syntenic regions were plotted using genoPlotR (v0.8.11) (59). The same approach was applied to *Verticillium alfalfae* (strain PD683) and *V. tricorpus* (strain PD593) (26), except that the 20-kb bin-size was used. Totally, 33 large syntenic regions were detected, covering 30.80 Mb and 32.48 Mb in *V. alfalfae* and *V. tricorpus*, respectively.

**Compartment analysis.** Genome compartmentation was calculated through principal-component analysis of the Hi-C contact matrix using module hicPCA in HiCExplorer (method set as "lieberman"). The contact matrices were binned to 20 kb and 100 kb for the data of CYR34 urediniospore and human (PRJNA375743) (60). The first three eigenvectors were computed. The eigenvector (usually the first or the second one) with the highest absolute correlation to that track was chosen as the one most likely corresponding to the compartments. In case the sign of the correlation coefficient is negative, the values of the eigenvector were flipped so that the regions with positive signs correspond to active regions. In addition, we manually validated the results by visualizing the correlations between eigenvectors and gene density/RNA-seq expression by plotting each chromosome. Different significance levels of gene density and expression level between "+" and "−" regions were tested using the one-tail Welch $t$ test under the assumption that the numbers in the PC+ regions are higher than those in the PC− regions (Fig. 5; see also Fig. S7 and S9 in the supplemental material).

**Reidentification of the *AvYr44-AvYr7-AvYr43-AvYrExp2* gene cluster.** Whole-genome sequencing and virulence phenotypes of 97 selfing progeny isolates were retrieved from previous study (29) and mapped to the CYR34 and 93-210 genomes. High-quality single nucleotide polymorphisms (SNPs) were filtered and used for quantitative trait locus (QTL) mapping of the *AvYr44-AvYr7-AvYr43-AvYrExp2* gene cluster as previously described (29).

**Data availability.** The raw sequencing reads are available in the NCBI database under accession number PRJNA807516 and SRA accessions SRX14387854 to SRX14387867. Assembled genomes and annotations generated in this study have been deposited to the NCBI database under accession number PRJNA807516, with genome accessions JANSJV000000000 for CYR34 and JANSJU000000000 for *P. striiformis* f. sp. *tritici* 93-210.

## SUPPLEMENTAL MATERIAL

Supplemental material is available online only.

**SUPPLEMENTAL FILE 1**, PDF file, 6 MB.

**SUPPLEMENTAL FILE 2**, MP4 file, 7.2 MB.

**SUPPLEMENTAL FILE 3**, XLSX file, 0.02 MB.

## ACKNOWLEDGMENTS

This work was financially supported by the National Key Research and Development Program (2021YFD1401000 and 2018YFD0200500), China Agriculture Research System (CARS-3), Agricultural Science and Technology Innovation Program (CAAS-ASTIP), and Epidemic Detection and Control of Crop Diseases and Insect Pests (2130108) to W. Chen and T. Liu; the National Natural Science Foundation of China (grant no. 31871906 to T. Liu; grant no. 31901833 to C. Xia); and the PhD Foundation of Southwest University of Science and Technology (no. 19zx7116) and Open Project of the State Key Laboratory for Biology of Plant Diseases and Insect Pests (SKLOF202113) to C. Xia.

The authors acknowledge the Kamiak high-performance computing clusters at Washington State University for providing computing resources. The authors are also grateful to the authors of the publicly available data used in this study. The authors particularly thank Peter M. Henry from USDA-ARS at Salinas, CA for sharing the *Fusarium oxysporum* f. sp. *fragariae* GL1381 and GL1080 genomes.

We declare no conflict of interest.

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
