## [Reviewer comments · Microbiology Spectrum]

Microbiology Spectrum

Folding features and dynamics of 3D genome architecture in plant fungal pathogens

Chongjing Xia, Liang Huang, Jie Huang, Hao Zhang, Ying Huang, Moussa Benhamed, Meinan Wang, Xianming Chen, Min Zhang, Taiguo Liu, and Wanquan Chen

Corresponding Author(s): Chongjing Xia, Chinese Academy of Agricultural Sciences

Review Timeline:

Submission Date:	July 8, 2022
Editorial Decision:	July 31, 2022
Revision Received:	September 1, 2022
Accepted:	September 18, 2022

Editor: Jing Han

Reviewer(s): Disclosure of reviewer identity is with reference to reviewer comments included in decision letter(s). The following individuals involved in review of your submission have agreed to reveal their identity: Xu Feng (Reviewer #1); Fei Sun (Reviewer #2); Ilya Flyamer (Reviewer #3)

Transaction Report:

DOI: <https://doi.org/10.1128/spectrum.02608-22>

July 31, 2022

Dr. Chongjing Xia
Chinese Academy of Agricultural Sciences
Institute of Plant Protection
State Key Laboratory for Biology of Plant Diseases and Insect Pests
Beijing, Beijing 100193
China

Re: Spectrum02608-22 (Folding features and dynamics of 3D genome architecture in plant fungal pathogens)

Dear Dr. Chongjing Xia:

Link Not Available

Sincerely,

Jing Han

Journals Department
Reviewer comments:

Reviewer #1 (Public repository details (Required)):

the NGS data should be deposited in a public repository.

Reviewer #1 (Comments for the Author):

In general, the manuscript is clearly written and I have only a few minor points for the author to consider.

Line 41 & 42, it was suggested that the regulation of gene activities might be independent of the changes of genome

organization. However, in line 52 & 53, the author suggest that 3D genome features might contribute to the regulatory mechanism of gene expression. Please clarify these conclusions.

Could the authors discuss possible reasons/factors that contribute to the formation of compartments in certain fungi, i.e. *Epichloe festucae*? Does this organism encode extra SMC family proteins compared to other fungi?

Minor comments.

Line 38, it should be "the fungi". 'the' is omitted in many sentences throughout the manuscript.

Line 55, different from?

Line 82, 'beside these model fungi', please check the use of "beside".

Line 91, gene express

Line 115, a brief introduction/description about the experiment setup is recommended, albeit it has been mentioned in the method part.

Line 126, please specify the "LTR elements and DNA elements"

Line 154, 'examined and plotted' should be changed to 'examine and plot'

Line 186, genomic locations

Line 194, Does this conclusion in the title apply to all studied fungal?

Line 249, in my understanding, the last paragraph of the Results section seems belonging to another story.

Line 261, together with data of other wheat rust fugal

Line 302, compartmentalization was not observed in all species.

Line 316, The low expression in peri-centromeric heterochromatin regions could also be a result, but not a driver for EHE formation.

Line 358 & Line 359, 'Shacking' should be changed to 'shaking'

Line 422, 'A genome'

Line 452, do you mean 'active or inactive'?

Reviewer #2 (Public repository details (Required)):

This study include NGS data.

Reviewer #2 (Comments for the Author):

In this study, Xia et al. employed Hi-C to reconstruct the chromatin interaction maps in *Puccinia striiformis* f. sp. *Tritici* (Pst), a plant pathogenic fungus that causes wheat rust disease. With these data, the authors discovered that wheat rust fungi adopt Rabl configuration and their genomes are organized into distinct globule models (fractal vs equilibrium) on different scales. By comparing the genome organization and gene expression between different growth stages, they found gene regulation is independent of 3D structure of the genome. Then the authors investigated if chromosome conformation at syntenic regions is conserved between species. Unexpectedly, distinct from mammalian genomes, 3D chromatin conformation is not conserved between filamentous fungi even at syntenic regions. The authors also discovered that A/B compartments seem not exist in wheat rust fungi, suggesting a very different genome organization model in these species. I found this manuscript is well organized and provides useful resources as well as solid conclusions. Below are some comments that in my opinion, may help improve this paper.

1. First paragraph in RESULTS. As a non-expert in wheat rust fungi, I had no idea what CYR34 or 93-210 was until I found some introduction to them in MATERIALS AND METHODS. It's better to move these words into the main text before presenting the data.

2. Fig 1. I understand no Pst genome had been successfully assembled before this study, so it's fundamental to describe the assembly of the 2 Pst genomes (CYR34 and 93-210) before presenting the Hi-C data. However, I don't think it's necessary to show Fig 1 in the main text. It's not related to the main theme of this study --- genome organization. Besides, panel A is difficult to read. It may be a good idea to move it into supplemental materials.

3. Paragraph starting at line 154. The 2009 Science paper by Liberman-Aiden (the first Hi-C paper in which globule models were explained) should be cited.

4. Fig 2d. The words inside each box are too small to read.

5. Fig 3d. This analysis is inappropriate and does not support the conclusion that "fold changes of gene expression in these two regions had no significant difference". The authors cherry-picked only the affected genes. It's very likely that there will be no significant difference if up-regulated genes in dis-similar regions are compared to up-regulated genes in conservation regions. I think the right way to do this is to 1) analyze all the genes in the dis-similar regions and see how their expression level changes

between different growth stages (i.e. whether or not these genes tend to be up- or down-regulated); 2) do the same to all the genes in the conservation regions and see if they display the same tendency as in the dis-similar regions.

6. The "Avirulence gene cluster" paragraph. The description is too general. Fig 6B and E, please use arrows to indicate the centromere regions.

7. There are a bunch of grammar mistakes in the manuscript. It's helpful to turn on the grammar check function in Microsoft Word.

Reviewer #3 (Public repository details (Required)):

All sequencing results are available via SRA. However I couldn't find the final genome assemblies. I think they are typically submitted to genbank? Or did I miss them somewhere?

Reviewer #3 (Comments for the Author):

In this paper the authors assembled genomes of two wheat rust fungi, using short and long read sequencing, and also Hi-C data. Then they used this Hi-C data to analyse the 3D organization of chromatin in these fungi, and to compare it to gene expression and synteny.

Overall this is an interesting study shedding light on some unique features of 3D organization in fungi. However I have a few important comments for the authors to consider regarding analysis and interpretation of Hi-C data.

1. First of all, when authors analyse the P(s) curves they only think in terms of fractal vs equilibrium globules, which is a highly simplified approach to chromatin structure. Do the authors think there is loop extrusion occurring in these species? From the flatter part in shorter distanced and a sharper drop later, I would expect so, this should be mentioned and discussed.

2. Why do inter-chromosomal contacts vary by distance? There are no distances, since regions are on different chromosomes there, so it is typically depicted as a single value.

3. Specifically the CYR34 germtube sample appears particularly interesting. To my knowledge, decay of contact frequency with the exponent of -2.7 has never been observed. This represents some highly unusual chromatin folding, unlikely to be any sort of globule even (certainly not equilibrium globule, as suggested by authors). Is there anything special going on with nuclear organization at this stage?

Note that on the line 156 there is a typo: $s^{-3/2}$ corresponds to an equilibrium globule, not fractal globule.

4. Related to the same figure, the 3D modelling performed by the authors is incorrect. The tool is they used is generally designed for single-cell Hi-C, not bulk Hi-C like here. Interestingly, the tool does allow to process bulk data, but requires an argument -p that the authors did not use. In general, generating 3D structure from population/bulk Hi-C doesn't make much sense in my opinion (since it's averaged from millions of different conformations), and I would simply remove this part.

5. Regarding compartment analysis, I see some some misconception there. First of all, the sign of the eigenvector on its own is meaningless. Typically, the compartment signal is obtained by ensuring positive correlation of the first (or another top) eigenvector with something that would correlate with "active" regions. For example, in mammals GC content works very well, since genes have higher GC content than genomic average. Gene density of RNA-seq signal should work well in any organism with compartments. Then the eigenvector that has the highest absolute correlation with that track is chosen as the one that most likely corresponds to compartments, and in case the sign of the correlation coefficient is negative, the values of the eigenvector are flipped, so this way the regions with positive signs correspond to active regions.

6. In case the compartments are very weak and absent, the eigenvectors don't pick up eu- vs heterochromatin. They would pick up some other major features of the contact map, very often the centromeres vs chromosome arms (as in this case), or sometimes perhaps left arm vs right arm... So I wouldn't draw any conclusions about the chromatin state based on the eigenvectors in case with no visible compartments (of if the eigenvector for some reason didn't pick out the compartment structure).

7. Would be interesting to see some speculations about why only one of the fungi analysed here displays clear compartments, but it's understandable if authors think there is not enough known to even think about it here.

Finally, the writing is completely understandable, but would benefit from proofreading to remove occasional typos or grammatical errors.

Staff Comments:

Preparing Revision Guidelines

Please return the manuscript within 60 days; if you cannot complete the modification within this time period, please contact me. If you do not wish to modify the manuscript and prefer to submit it to another journal, please notify me of your decision immediately so that the manuscript may be formally withdrawn from consideration by Microbiology Spectrum.

In this manuscript, Xia *et al* generate two chromosome-level genome assemblies for a plant fungi, *pst*, using Hi-C assisted scaffolding pipeline and investigated the chromosome conformations of this organism and a few other fungi.

In general, the manuscript is clearly written and I have only a few minor points for the author to consider.

Line 41 & 42, it was suggested that the regulation of gene activities might be independent of the changes of genome organization. However, in line 52 & 53, the author suggest that 3D genome features might contribute to the regulatory mechanism of gene expression. Please clarify these conclusions.

Could the authors discuss possible reasons/factors that contribute to the formation of compartments in certain fungi, i.e. *Epichloe festucae*? Does this organism encode extra SMC family proteins compared to other fungi?

Minor comments.

Line 38, it should be “the fungi”. ‘the’ is omitted in many sentences throughout the manuscript.

Line 55, different from?

Line 82, ‘beside these model fungi’, please check the use of “beside”.

Line 91, gene express

Line 115, a brief introduction/description about the experiment setup is recommended, albeit it has been mentioned in the method part.

Line 126, please specify the “LTR elements and DNA elements”

Line 154, ‘examined and plotted’ should be changed to ‘examine and plot’

Line 186, genomic locations

Line 194, Does this conclusion in the title apply to all studied fungal?

Line 249, in my understanding, the last paragraph of the Results section seems belonging to another story.

Line 261, together with data of other wheat rust fugal

Line 302, compartmentalization was not observed in **all** species.

Line 316, The low expression in peri-centromeric heterochromatin regions could also be a result, but not a driver for EHE formation.

Line 358 & Line 359, ‘Shacking’ should be changed to ‘shaking’

Line 422, ‘A genome’

Line 452, do you mean ‘active or inactive’?

Reviewer comments:

We thank all reviewers for their constructive comments and suggestions to improve the quality and clarity of the manuscript. Below are our responses to each point of the comments or suggestions.

Reviewer #1 (Public repository details (Required)):

the NGS data should be deposited in a public repository.

Response: Thanks. To make the data repository clear, we have added accession numbers in the "Data availability" section.

Reviewer #1 (Comments for the Author):

In general, the manuscript is clearly written and I have only a few minor points for the author to consider.

Response: Thanks for the positive feedback. Every suggestion is highly appreciated and considered in our revision.

Line 41 & 42, it was suggested that the regulation of gene activities might be independent of the changes of genome organization. However, in line 52 & 53, the author suggest that 3D genome features might contribute to the regulatory mechanism of gene expression. Please clarify these conclusions.

Response: Thanks for pointing out this misleading. To clarify this, we changed the statement as "There might be distinct regulatory mechanisms of gene expression that are independent of changes of chromatin organization during developmental stages of rust fungi." (Lines 57-58)

Could the authors discuss possible reasons/factors that contribute to the formation of compartments in certain fungi, i.e. *Epichloe festucae*? Does this organism encode extra SMC family proteins compared to other fungi?

Response: We obtained the annotated proteomes of 8 fungi, and searched for the presence of PF02463, the domain at the N terminus of SMC proteins. Unfortunately, it seems the numbers of SMC proteins do not have much difference in the examined fungi.

In fact, the mechanism underlying the formation of compartments is still unclear. The most updated idea is that formation of compartment is mediated by homotypic interactions among genome regions with the same transcriptional state, and then generated phase-separated structures. Based on this, we could only provide some thinking on possible reasons why compartment is present in certain fungi but not others. We have added sentences to discuss the point (Lines 348-360).

Minor comments.

Line 38, it should be "the fungi". 'the' is omitted in many sentences throughout

the manuscript.

Response: Thanks. We have added “the” when appropriate throughout the manuscript.

Line 55, different from?

Response: We have changed “different with” to “different from” (Line 59).

Line 82, 'beside these model fungi', please check the use of "beside".

Response: Thanks. We have changed “beside” to “Aside from” (Line 87).

Line 91, gene express

Response: We have changed “gene express” to “gene expression” (Line 97).

Line 115, a brief introduction/description about the experiment setup is recommended, albeit it has been mentioned in the method part.

Response: Thanks for the suggestion. We have added a short paragraph (Lines 122-137).

Line 126, please specify the "LTR elements and DNA elements"

Response: Thanks, we have modified the sentence (Line 149-152). In addition, we specified the most two abundant sub-family under LTR and DNA element families. We updated this information in Supplementary Table 2.

Line 154, 'examined and plotted' should be changed to 'examine and plot'

Response: changed (Line 180).

Line 186, genomic locations

Response: Changed (Line 225).

Line 194, Does this conclusion in the title apply to all studied fungal?

Response: Thanks for the comment. We have modified the sub-heading by adding “in the examined fungi” (Line 233). We do admit that only very limited number of fungi have Hi-C data available, so limited our examination on more fungi. Therefore, we added a notice at the end of this paragraph, “Further studies will be needed to examine this independence among diverse fungi when high-quality Hi-C data are available.” (Line 247-248).

Line 249, in my understanding, the last paragraph of the Results section seems belonging to another story.

Response: Thanks for the comment. We sort of agree with your point, but we intended to use the analyses of our previously identified avirulence gene cluster as an example to show how chromosome-level genome and its 3D organization provide more in depth understanding of genomic environment of pathogen virulence genes.

Line 261, together with data of other wheat rust fugal

Response: Changed (Line 303).

Line 302, compartmentalization was not observed in all species.

Response: Changed (Line 345).

Line 316, The low expression in peri-centromeric heterochromatin regions could also be a result, but not a driver for EHE formation.

Response: Thanks for the comment. We have revised the sentence (Line 374).

Line 358 & Line 359, 'Shacking' should be changed to 'shaking'

Response: Corrected (Lines 415-416).

Line 422, 'A genome'

Response: Changed (Line 480)

Line 452, do you mean 'active or inactive'?

Response: Changed (Line 515)

Reviewer #2 (Public repository details (Required)):

This study include NGS data.

Response: Thanks. To make the data repository clear, we have added accession numbers in the "Data availability" section.

Reviewer #2 (Comments for the Author):

In this study, Xia et al. employed Hi-C to reconstruct the chromatin interaction maps in *Puccinia striiformis* f. sp. *Tritici* (Pst), a plant pathogenic fungus that causes wheat rust disease. With these data, the authors discovered that wheat rust fungi adopt Rabl configuration and their genomes are organized into distinct globule models (fractal vs equilibrium) on different scales. By comparing the genome organization and gene expression between different growth stages, they found gene regulation is independent of 3D structure of the genome. Then the authors investigated if chromosome conformation at syntenic regions is conserved between species. Unexpectedly, distinct from mammalian genomes, 3D chromatin conformation is not conserved between filamentous fungi even at syntenic regions. The authors also discovered that A/B compartments seem not exist in wheat rust fungi, suggesting a very different genome organization model in these species. I found this manuscript is well organized and provides useful resources as well as solid conclusions. Below are some comments that in my opinion, may help improve this paper.

Response: Thanks for your positive evaluation. Your suggestions are highly

appreciated and considered in our revision.

1. First paragraph in RESULTS. As a non-expert in wheat rust fungi, I had no idea what CYR34 or 93-210 was until I found some introduction to them in MATERIALS AND METHODS. It's better to move these words into the main text before presenting the data.

Response: We have added a short introduction paragraph before presenting the results (Lines 122-137).

2. Fig 1. I understand no Pst genome had been successfully assembled before this study, so it's fundamental to describe the assembly of the 2 Pst genomes (CYR34 and 93-210) before presenting the Hi-C data. However, I don't think it's necessary to show Fig 1 in the main text. It's not related to the main theme of this study --- genome organization. Besides, panel A is difficult to read. It may be a good idea to move it into supplemental materials.

Response: The two chromosome-level genomes are major results of this study and valuable resources for studying *Pst* study. They are the basis for the genome organization analyses. We have improved the resolution in Fig. 1 in the revised manuscript.

3. Paragraph starting at line 154. The 2009 Science paper by Liberman-Aiden (the first Hi-C paper in which globule models were explained) should be cited.

Response: Thanks for the suggestion. We have added a sentence about this paper (Line 182).

4. Fig 2d. The words inside each box are too small to read.

Response: We changed the font size to make them readable. In addition, we provided the high-resolution figures in the revised manuscript.

5. Fig 3d. This analysis is inappropriate and does not support the conclusion that "fold changes of gene expression in these two regions had no significant difference". The authors cherry-picked only the affected genes. It's very likely that there will be no significant difference if up-regulated genes in dis-similar regions are compared to up-regulated genes in conservation regions. I think the right way to do this is to 1) analyze all the genes in the dis-similar regions and see how their expression level changes between different growth stages (i.e. whether or not these genes tend to be up- or down-regulated); 2) do the same to all the genes in the conservation regions and see if they display the same tendency as in the dis-similar regions.

Response: Thanks for the comment. We re-analyzed the data, following your suggestion, to see the overall changes of expression of all the genes in the dis-similar regions and in the conservation regions. We still use $\log_2(\text{fold change})$ as a function of expression changes. The gene is up-regulated when $\log_2(\text{fold change}) > 0$, while down-regulated when $\log_2(\text{fold change}) < 0$. The results are shown in revised Fig 3d.

When calculated using all genes, the mean values of $\log_2(\text{fold change})$ are -0.448 and -0.547 for genes in the dis-similar and conservation regions, respectively. The Welch Two Sample t-test indicated there is no significant difference of expression changes between these two regions. This further confirmed our conclusion. We have revised this part (Lines 218-224).

6. The "Avirulence gene cluster" paragraph. The description is too general. Fig 6B and E, please use arrows to indicate the centromere regions.

Response: We have modified Fig 6 and the sentences in the text (Lines 135-137).

7. There are a bunch of grammar mistakes in the manuscript. It's helpful to turn on the grammar check function in Microsoft Word.

Response: Thanks for the suggestion. We have corrected the mistakes throughout the manuscript.

Reviewer #3 (Public repository details (Required)):

All sequencing results are available via SRA. However I couldn't find the final genome assemblies. I think they are typically submitted to genbank? Or did I miss them somewhere?

Response: Thanks. To make the data repository clear, we have added accession numbers in the "Data availability" section.

Reviewer #3 (Comments for the Author):

In this paper the authors assembled genomes of two wheat rust fungi, using short and long read sequencing, and also Hi-C data. Then they used this Hi-C data to analyse the 3D organization of chromatin in these fungi, and to compare it to gene expression and synteny.

Overall this is an interesting study shedding light on some unique features of 3D organization in fungi. However I have a few important comments for the authors to consider regarding analysis and interpretation of Hi-C data.

Response: Thanks for the positive review. Your comments and suggestions are highly appreciated and considered in our revision.

1. First of all, when authors analyse the $P(s)$ curves they only think in terms of fractal vs equilibrium globules, which is a highly simplified approach to chromatin structure. Do the authors think there is loop extrusion occurring in these species? From the flatter part in shorter distanced and a sharper drop later, I would expect so, this should be mentioned and discussed.

Response: The fractal and equilibrium globules are two basic models to describe chromosome structures. One of our objectives is to describe the overall organization feature of fungi at the chromosome level, instead of finer scales. We noticed the flatter

parts in the $P(s)$ curves contained contacts within 1 kb (Fig 2d), but currently, could not explain this yet. These contacts might be not resulted from the loop extrusion, since the fungi are lack of cohesin, the observed organization could not be mediated by loop extrusion which needs cohesin. In addition, our unpublished data did not detect prominent loop structures in the rust fungi. So, we prefer to leave the finer scale of the organization for further studies that may need higher resolution Hi-C data. We have modified this part (Lines 348-360)

2. Why do inter-chromosomal contacts vary by distance? There are no distances, since regions are on different chromosomes there, so it is typically depicted as a single value.

Response: To calculate inter-chromosomal contact probability, we combined the two chromosomes as a super-chromosome. Then the contact probability was calculated between the two loci, one from one chromosome and one from the other chromosome. We have added this information in Methods (Lines 494-496).

3. Specifically the CYR34 germtube sample appears particularly interesting. To my knowledge, decay of contact frequency with the exponent of -2.7 has never been observed. This represents some highly unusual chromatin folding, unlikely to be any sort of globule even (certainly not equilibrium globule, as suggested by authors). Is there anything special going on with nuclear organization at this stage?

Response: The Germ tube is a growth structure produced by germinated spores. This structure grows rapidly, in terms of elongation, in order to reach infection sites (the wheat stomata) for growing into plant tissue. So, the germ-tube cells are highly active mitotic cells. Therefore, we speculate what was captured by the Hi-C data was the transient feature of chromosome state from germ-tube cells, instead of chromatin contacts as from the dormant urediniospores. This could be supported by the Hi-C heatmap of germ tubes (Fig 2a) in which the inter-chromosome contacts were significantly diminished (9.7% in germtubes vs 29.93% in urediniospores; Table S3). We reason that the exponent of -2.7 indicates the contact probability of two loci decreases sharply as genomic distance increases. In other words, there are less contacts between loci with large distances (> 20 kb), which suggests a condenser organization, and therefore is consistent with the chromosome state. The middle panel of Fig 3b could be an example, e.g., the normalized contact heatmap of germtube is devoid of long-range contacts (light color), comparing to the heatmap of urediniospore (deep color). We have modified the part to discuss this (Lines 191-199)

Note that on the line 156 there is a typo: $s^{-3/2}$ corresponds to an equilibrium globule, not fractal globule.

Response: Thanks, the typo has been corrected. (Line 182)

4. Related to the same figure, the 3D modelling performed by the authors is

incorrect. The tool is they used is generally designed for single-cell Hi-C, not bulk Hi-C like here. Interestingly, the tool does allow to process bulk data, but requires an argument -p that the authors did not use. In general, generating 3D structure from population/bulk Hi-C doesn't make much sense in my opinion (since it's averaged from millions of different conformations), and I would simply remove this part.

Response: Thanks for pointing out this. To address this issue, we have added sentence in the revised manuscript (Lines 499-504). To model the 3D structure, there were two steps. First, Hi-C reads were processed to generate contacts using program nuc_process. Second, nuc_dynamics was used to calculate 3D coordinates from contact maps. We noticed these programs were generally designed for single-cell Hi-C. Actually, we used the argument -p, following the developer's recommendation, to process multi-cell Hi-C in nuc_process. The full arguments for nuc_process running was: -g genome.fasta -f genome.fasta -re1 Mbol -n 20 -r 3 -o cyr34 -or report -b /path/to/bowtie2 -p -k -sam -v. In addition, when using nuc_dynamics to construct the 3D genome, we used the root mean square deviation (RMSD) as an indicator to evaluate the consistency of the structures in the ensemble, following the developer's recommendation. In fact, the constructed 3D structures presented in this study all had RMSD values less than 2.0 were presented in this study. (please note that, for structures with RMSD > 2.0, e.g. at 5 kb resolution, we presented the lower resolution with RMSD <2.0, e.g. at 10 kb resolution).

5. Regarding compartment analysis, I see some some misconception there. First of all, the sign of the eigenvector on its own is meaningless. Typically, the compartment signal is obtained by ensuring positive correlation of the first (or another top) eigenvector with something that would correlate with "active" regions. For example, in mammals GC content works very well, since genes have higher GC content than genomic average. Gene density of RNA-seq signal should work well in any organism with compartments. Then the eigenvector that has the highest absolute correlation with that track is chosen as the one that most likely corresponds to compartments, and in case the sign of the correlation coefficient is negative, the values of the eigenvector are flipped, so this way the regions with positive signs correspond to active regions.

Response: Thanks for the comment. Indeed, this was exactly what we did for compartment analysis. To make the analyses clear, we modified the sentences in the method part (Lines 532-537).

6. In case the compartments are very weak and absent, the eigenvectors don't pick up eu- vs heterochromatin. They would pick up some other major features of the contact map, very often the centromeres vs chromosome arms (as in this case), or sometimes perhaps left arm vs right arm... So I wouldn't draw any conclusions about the chromatin state based on the eigenvectors in case with no visible compartments (of if the eigenvector for some reason didn't pick out the compartment structure).

Response: Thanks for the comment. Since the eigenvectors do not associate with compartments, we attempted to find other major features, and found that the eigenvectors could pick up the two chromosome arms and the centromere. Considering that the eigenvectors are usually associated with expression (active vs inactive compartments) in high eukaryotes, then we speculated the observed organization in fungi might be associated with chromatin state, e.g., eu- vs heterochromatin. While the data is still limited, we provided one example of well-studied fungus *Neurospora crassa* to support this hypothesis. In this fungus, the centromere regions picked up by eigenvectors are enriched in heterochromatin-associated H3K9me3 modification (supplementary Fig 10). To highlight this speculation, we revised the part in Lines 361-366.

7. Would be interesting to see some speculations about why only one of the fungi analysed here displays clear compartments, but it's understandable if authors think there is not enough known to even think about it here.

Response: We provided our speculation in the revised manuscript (Lines 348-360).

Finally, the writing is completely understandable, but would benefit from proofreading to remove occasional typos or grammatical errors.

Response: Thanks. We have carefully checked and corrected the typos and grammatical errors.

September 18, 2022

Dr. Chongjing Xia
Chinese Academy of Agricultural Sciences
Institute of Plant Protection
State Key Laboratory for Biology of Plant Diseases and Insect Pests
Beijing, Beijing 100193
China

Re: Spectrum02608-22R1 (Folding features and dynamics of 3D genome architecture in plant fungal pathogens)

Dear Dr. Chongjing Xia:

Your manuscript has been accepted, and I am forwarding it to the ASM Journals Department for publication. You will be notified when your proofs are ready to be viewed.

Sincerely,

Jing Han
Editor, Microbiology Spectrum

Journals Department
Supplemental Movie 1: Accept
Supplemental Dataset: Accept
Supplemental Material: Accept